# Regulatory Role of *Meox1* in Muscle Growth of *Sebastes schlegelii*

**DOI:** 10.3390/ijms25094871

**Published:** 2024-04-29

**Authors:** Weihao Song, Xiaotong Liu, Kejia Huang, Jie Qi, Yan He

**Affiliations:** MOE Key Laboratory of Marine Genetics and Breeding, College of Marine Life Sciences, Ocean University of China, Qingdao 266003, China; songweihao8@163.com (W.S.); lxt4574@stu.ouc.edu.cn (X.L.); 15071255840@163.com (K.H.); qijie@ouc.edu.cn (J.Q.)

**Keywords:** *SsMeox1*, muscle growth, growth-specific muscle stem cell, cell cycle, *Sebastes schlegelii*

## Abstract

*Meox1* is a critical transcription factor that plays a pivotal role in embryogenesis and muscle development. It has been established as a marker gene for growth-specific muscle stem cells in zebrafish. In this study, we identified the *SsMeox1* gene in a large teleost fish, *Sebastes schlegelii*. Through in situ hybridization and histological analysis, we discovered that *SsMeox1* can be employed as a specific marker of growth-specific muscle stem cells, which originate from the somite stage and are primarily situated in the external cell layer (ECL) and myosepta, with a minor population distributed among muscle fibers. The knockdown of *SsMeox1* resulted in a significant increase in *Ccnb1* expression, subsequently promoting cell cycle progression and potentially accelerating the depletion of the stem cell pool, which ultimately led to significant growth retardation. These findings suggest that *SsMeox1* arrests the cell cycle of growth-specific muscle stem cells in the G2 phase by suppressing *Ccnb1* expression, which is essential for maintaining the stability of the growth-specific muscle stem cell pool. Our study provides significant insights into the molecular mechanisms underlying the indeterminate growth of large teleosts.

## 1. Introduction

Muscle growth relies on an increase in both muscle fiber volume (hypertrophy) and the number of muscle fibers (hyperplasia) [1,2]. In amniotes, the number of muscle fibers is determined before birth, and postnatal muscle growth is entirely driven by the hypertrophy of the existing muscle fibers [3]. Recent studies have revealed that muscle growth in zebrafish (*Danio rerio*) is determinate, with minimal muscle fiber hyperplasia occurring after the juvenile stage [4]. In contrast, large teleost fish exhibit simultaneous muscle hyperplasia and hypertrophy throughout their life cycle [2]. Thus, the muscle development of large teleosts displays indeterminate growth, a characteristic that is distinctly different from that of amniotes and small teleosts.

Muscle stem cells drive indeterminate skeletal muscle growth, and they are still active in the adult stage of large teleosts. Two populations of muscle stem cells have been identified in zebrafish [5,6]. Satellite cells, located between the myolemma and basement membrane, are activated and proliferated in response to muscle injury to repair muscle fibers [7,8]. Quiescent satellite cells are in G0 arrest [9,10,11]. Another population is growth-specific muscle stem cells, which contribute to the proliferation of fish muscle fibers. They are mainly located in the ECL (external cell layer) area and the muscular septum [5]. The ECL is produced by the unique segmental turnover activity of the teleost, during the embryonic stage [12,13]. ECLs and growth-specific muscle stem cells have not been found in amniotes and may be key factors in the generation of different muscle growth strategies. It is important to know that the population of satellite cells generally remains highly heterogeneous, which is favorable for extreme stimulation [14,15,16]. However, growth-specific muscle stem cells tend to be consistent.

In zebrafish, *Meox1* has been demonstrated to suppress the expression of *CCNA2* and *CcnB1*, leading to cell cycle arrest of growth-specific muscle stem cells in the G2 phase [5]. This finding highlights the crucial role of *Meox1* in regulating the proliferation and differentiation of these stem cells, which are essential for muscle growth and development. Intriguingly, while *Meox1*-deficient humans and mice do not exhibit overt muscle development defects, they do present with cervical skeletal malformations, a phenotype also observed in zebrafish [17,18,19,20]. This discrepancy suggests that the role of *Meox1* in muscle development may vary across species, with potentially distinct mechanisms governing muscle growth in different vertebrate lineages. Although the importance of *Meox1* and growth-specific muscle stem cells in secondary myogenic growth has been well-established in zebrafish [5,21], their role in large teleost fish exhibiting indeterminate growth remains largely unexplored. 

*Sebastes schlegelii*, an economically important marine fish species in northern China, exhibits an indeterminate muscle growth pattern [22]. In this study, we employed *Meox1* as a marker gene to localize and trace the origin of growth-specific muscle stem cells in *S. schlegelii*. Furthermore, we investigated the role of *Meox1* in regulating muscle growth in *S. schlegelii* using a knockdown approach by RNAi, which resulted in a 53% decrease in *Meox1* expression at 40 days post-treatment. Our findings provide insights into the unlimited muscle growth potential of large teleost fish and lay the theoretical foundation for the development of targeted strategies to promote muscle growth in fish.

## 2. Results

### 2.1. Gene Structure, Synteny Analysis, and Evolution of Meox1

The CDS sequence of *SsMeox1* and its gene structure were identified based on the genomic and transcriptomic data of *S. schlegelii*. The gene structure of *SsMeox1* conforms to the typical pattern observed in most teleost fishes, comprising three exons and two introns (Figure 1A). Gene synteny analysis revealed *Sost* and *Etv4* as neighboring genes flanking *SsMeox1*, a conservation pattern observed across various investigated species (Figure 1B). Subsequently, a phylogenetic analysis based on amino acid sequences was conducted to ascertain the evolutionary placement of *SsMeox1* (Figure 1C). The results delineated *Meox1* and its paralogous gene *Meox2* into distinct branches, indicating their independent evolutionary trajectories. Notably, a comparative analysis unveiled a relatively low homology between *SsMeox1* and *Meox1* genes of small teleosts, such as *Poecilia reticulata* and *Oryzias latipes*. These findings suggest that *Meox1* predates the third round of whole-genome duplication (3R-WGD), affirming its status as an ancestral gene. Furthermore, the observed distinctions in homology hint at potential functional divergence of *Meox1* between small and large teleost species during independent evolutionary events.

### 2.2. Protein Structure of Meox1

The alignment of *Meox1* amino acid sequences revealed a highly conserved homeobox domain (Figure 2A). The homeobox domain is well-known for its role as a DNA-binding domain, defining proteins possessing it as transcription factors (TFs) [23]. However, notable distinctions were observed in the amino acid sequences of *Meox1* genes between teleosts and amniotes, suggesting potential differences in their biological functions. Utilizing the SWISS-MODEL, we constructed a 3D protein structure of *SsMeox1*, unveiling three α-helices at the N-terminus (Figure 2B). The presence of a peptide motif with a helix–loop–helix–turn–helix structure underscores the significance of the homeobox domain, emphasizing its association with DNA binding [23].

### 2.3. Identification of Growth-Specific Muscle Stem Cells in S. schlegelii

To further explore the role of *SsMeox1* in muscle growth in *S. schlegelii*, we first examined the tissue expression patterns of *SsMeox1* based on transcriptomic data. It was found that *SsMeox1* exhibited the highest expression in muscle tissue (Figure 3A). Although high expression was also detected in the gills, it was noted that most RNA in the gills originated from muscle and blood. Interestingly, the expression of *SsMeox1* in muscle tissue was higher in females compared to males, potentially elucidating the larger body size of female individuals at the same age. Additionally, the expression of *SsMeox1* in muscle tissue decreased gradually during postembryonic development (Figure 3B), which appears to be positively correlated with the growth rate of *S. schlegelii*.

Previous studies have shown that *Meox1* specifically marks growth-specific muscle stem cells during embryonic development and early development stages in zebrafish [5]. However, the existence of growth-specific muscle stem cells in the muscles of large teleosts remains unclear. To address this, growth-specific muscle stem cells were identified through hematoxylin–eosin (HE) staining and in situ hybridization (ISH) in *S. schlegelii* muscle tissue. A distinct class of monocytes was discovered in the muscular septum, the ECL region on the lateral axis of the body, and between muscle fibers (Figure 3C–C″′). Based on these results, these identified monocytes were found not to be muscle satellite cells. As no blood vessels were observed nearby, their classification as blood cells could also be ruled out. To further characterize these cells, the previously identified growth-specific stem cell marker gene, *Meox1,* was utilized for ISH to pinpoint their location in *S. schlegelii* (Figure 3D–E″). The signals aligned with the location of the identified monocytes, confirming their identity as growth-specific muscle stem cells.

### 2.4. Tracing the Embryonic Origin of Growth-Specific Muscle Stem Cells in S. schlegelii

We then conducted whole-mount in situ hybridization (WISH) to detect *SsMeox1* mRNA signaling at different embryonic stages, including the blastula, gastrula, somite stage, and pre-hatching stage to trace the origin of growth-specific muscle stem cells. Our results showed that the initiation of *SsMeox1* signaling concurred with the onset of the somite stage, while no discernible signal was evident in pre-somite stage embryos (Figure 4A–D′). This result is consistent with the transcriptome analysis, which demonstrated minimal *SsMeox1* expression preceding the somite stage (Figure 3B).

### 2.5. SsMeox1 Knockdown Hindered the Growth of S. schlegelii

To elucidate the mechanism underlying the influence of *SsMeox1* on muscle growth, we employed a knockdown approach in juvenile *S. schlegelii,* which was achieved via feeding a recombinant strain capable of producing *SsMeox1*-targeted dsRNA. Although this method has been extensively documented for gene knockdown in organisms such as elegans and insects, its validation in vertebrates remains unexplored [24,25,26,27]. The engineered HT115 strain demonstrated the capability to produce dsRNA fragments of the correct RNA size (Figure 5A) and, notably, the expression of *SsMeox1* in muscle was significantly suppressed by 53% at 40 days and 28% at 60 days upon application of the knockdown system to juvenile *S. schlegelii* (Figure 5B). Furthermore, we compared the body length across different treatment periods following *SsMeox1* knockdown. Remarkably, the body length of the *SsMeox1* knockdown group was significantly reduced compared to the control group after 60 days (Figure 5D), specifically, 36.28 ± 3.45 cm vs. 40.22 ± 3.66 cm at 60 days, and 53.70 ± 4.47 cm vs. 63.46 ± 3.53 cm at 120 days. Intriguingly, this disparity in body length was further accentuated at 60 days post-termination of the knockdown treatment (Figure 5C,D). Considering the spatial expression pattern of *SsMeox1*, we postulate that *SsMeox1* knockdown during the early stage might diminish the reservoir of muscle stem cells. Collectively, our findings suggest that *SsMeox1* knockdown exerts a negative influence on muscle growth, potentially achieved through the regulation of growth-specific muscle stem cell numbers.

### 2.6. Knockdown of SsMeox1 Expression Inhibited Enlargement of Muscle Fibers

Muscle fibers contribute significantly to the volume of muscle tissue. Thus, the cross-sectional area of all muscle fibers of the penultimate and antepenultimate myotome in the upper part of ECL was measured to assess the specific impact of *SsMeox1* on muscle growth (Figure 6A,A′). We found that the average fiber area in the *SsMeox1* knockdown group was significantly smaller than that in the control group after 60 days of treatment, which was consistent with the trend of the body length data (Figure 6B). Subsequently, we categorized all fibers into small fibers (<300 μm^2^), medium fibers (300–1000 μm^2^), and large fibers (>1000 μm^2^), manually. The *SsMeox1* knockdown group exhibited a higher proportion of medium-sized fibers compared to the control group after 30 days of treatment, while in subsequent developmental stages, the control group showed a higher proportion of medium and large fibers (Figure 6C). Our findings indicate that *SsMeox1* knockdown appears beneficial for early muscle development of *S. schlegelii*, suggesting that *SsMeox1* deficiency may activate lineage commitment of growth-specific muscle stem cells. However, a prolonged *SsMeox1* deficiency led to the depletion of the growth-specific muscle stem cell pool, impairing the driving force of muscle development.

### 2.7. SsMeox1 Knockdown Resulted in Increased Expression of Cell Cycle-Related Genes

To explore how *SsMeox1* affects the molecular mechanism of muscle development, samples from the 40-day *SsMeox1* knockdown group with the most significant knockdown effect were collected for transcriptome sequencing (hereinafter referred to as G-40D). The analysis of differentially expressed genes (DEGs) of G-40D samples yielded a considerable number of DEGs (Appendix A), and all DEGs were categorized using gene ontology (GO) and the Kyoto Encyclopedia of Genes and Genomes (KEGG) enrichment analysis. We observed that 11 terms are directly involved in cell cycle progression (Figure 7A). Moreover, KEGG enrichment analysis indicated that 53 DEGs were enriched in the cell cycle pathway, which exhibited the smallest *q*-value among all the pathways analyzed. This finding suggests that *SsMeox1* knockdown profoundly affects the cell cycle of growth-specific muscle stem cells, potentially resulting in a lack of motivation for subsequent muscle development. The initiation or arrest of the cell cycle is regulated by families such as cyclin, cyclin-dependent kinase (CDK), and cyclin-dependent kinase inhibitor (CDKI) [28,29]. Interestingly, the expression levels of these genes were all upregulated with more than a 2-fold change following *SsMeox1* knockdown. This finding indicates that *SsMeox1* knockdown can stimulate growth-specific muscle stem cells to enter the cell cycle.

### 2.8. SsMeox1 Inhibited the Expression of CcnB1 and Promoted the Expression of P21

Although a plethora of candidate genes have been identified, precise regulatory interactions with *SsMeox1* remain elusive. Utilizing the JASPAR database [30], we conducted a preliminary screening of the 12 genes referred to above, revealing eight genes harboring putative high-affinity binding sites within their 5’UTRs for potential interaction with *SsMeox1*. Subsequently, we employed a dual luciferase system for further validation. It is confirmed that the transcription of *CcnB1* can be repressed by *SsMeox1* directly (Figure 8A), consistent with the findings from the transcriptomic analyses. Intriguingly, *SsMeox1* exhibited a contrasting regulatory effect on *CDKN1A* (*p21*), as evidenced by upregulation in the dual luciferase reporter system (Figure 8C), contradicting the transcriptome data. Prior investigations implicated mouse *Meox1* in the induction of *p21* expression via DNA-independent mechanisms, a phenomenon not replicated in our experiment [31]. Notably, *SsMeox1* did not exert discernible regulatory effects on the transcription of the remaining six genes.

## 3. Discussion

*Meox1*, a transcription factor, has been extensively studied for its pivotal role in somite development [19]. While earlier investigations have mainly focused on its involvement in somitogenesis during embryonic stages. For example, *Meox1* null mutations in mice have been linked to mild defects in sclerotome-derived vertebral and rib bones [32]. *Meox1* knockdown can impede the differentiation of smooth muscle cells in pluripotent murine C3H10T1/2 cells [33]. Recently, emerging research has highlighted *Meox1*’s role in the normal cell cycle of muscle stem cells, which is crucial for muscle growth in zebrafish [5]. Thus, it is plausible to speculate that *Meox1* may also significantly contribute to muscle growth in other teleost species. 

We initially examined the expression pattern of *SsMeox1* in *S. schlegelii* tissues, utilizing previously published transcriptome data [22] and observed significant *SsMeox1* expression in muscle tissue, particularly heightened in females, which potentially elucidates sexual dimorphism in muscle growth. In contrast to zebrafish, where *Meox1* expression is confined to embryonic development stages [5], *SsMeox1* expression persists postnatally in *S. schlegelii*, implying a sustained impact on muscle growth. Considering the morphological disparities between *S. schlegelii* and zebrafish, this distinction appears reasonable.

Of note, previous studies have identified *Meox1* as a target for cancer stem cell treatment [34]; yet, its role in marking growth-specific stem cells in zebrafish was only recently reported [5]. ISH and WISH in *S. schlegelii* revealed *SsMeox1* expression in a population of monocytes primarily located in the ECL region along the lateral body axis, thus also establishing *SsMeox1* as a marker for growth-specific muscle stem cells in *S. schlegelii*. These findings significantly contribute to our understanding of muscle growth and offer a novel marker gene for growth-specific muscle stem cell research in *S. schlegelii*.

*Meox1* has recently been implicated in the ‘clonal drift’ of a long-lived stem cell pool in zebrafish [5]. Loss of *Meox1* activity correlates with reduced stem cell populations in the ECL region [21], ultimately leading to abnormal muscle growth or deficiency in muscle regeneration. To further elucidate *Meox1*’s role in muscle growth, we conducted *SsMeox1* knockdown experiments in juvenile *S. schlegelii* postnatally for 60 days. Remarkably, significant growth retardation was observed in the *SsMeox1* knockdown group compared to the control group, such as notable reductions in body length (e.g., 36.28 ± 3.45 cm vs. 40.22 ± 3.66 cm at 60 days), affirming *SsMeox1*’s association with muscle growth in *S. schlegelii*. Based on the spatial expression patterns of *SsMeox1*, we infer that *SsMeox1* knockdown negatively impacts the maintenance of growth-specific muscle stem cells, thereby inhibiting normal muscle growth. This effect mirrors the sarcopenia observed due to diminished satellite cell function [35]. Notably, this difference becomes more pronounced at 120 days after treatment, underscoring the long-lasting impact of *SsMeox1* knockdown on muscle growth.

The muscle growth patterns of *S. schlegelii* involve both hyperplasia and hypertrophy [36], with hypertrophy relying on the cell fusion of growth-specific stem cells to existing muscle fibers to enlarge the muscle fiber cross-sectional area [6]. We compared the proportion of muscle fibers with different cross-sectional areas between the *SsMeox1* knockdown group and the control group. Interestingly, we observed a significant difference 60 days after *SsMeox1* knockdown, but not at other stages, which may seem puzzling at first glance. Previous research indicates that hypertrophy becomes increasingly important for muscle growth between 30 to 100 days after birth [37]. Hence, we propose a hypothesis that, in the 60 days of *SsMeox1* knockdown, the remaining growth-specific muscle stem cells can still support muscle growth. However, by 60 days, the depletion of growth-specific muscle stem cells due to *SsMeox1* knockdown can no longer suffice to sustain hypertrophy, leading to a higher ratio of small muscle fibers. These results suggest that *SsMeox1* may play a crucial role in regulating muscle growth, particularly concerning hypertrophy, which is associated with growth-specific muscle stem cells.

To better understand the underlying mechanism, we conducted an analysis of DEGs and performed GO and KEGG enrichment analysis at 40 and 60 days, individually. At 40 days, we identified a number of DEGs associated with terms such as the regulation of the G2/M transition of the mitotic cell cycle and pathways like DNA replication, which are linked to the mitotic cell cycle [38]. This suggests that *SsMeox1* plays a crucial role in cell cycle control in *S. schlegelii*, akin to its function in zebrafish [5]. This indicates that the impact of *SsMeox1* knockdown on stem cells may have been determined by 40 days. 

*Meox1* and *Ccnb1* collectively regulate the cell cycle of growth-specific muscle stem cells, potentially by arresting them in the G2 phase, where cells are more responsive to growth signals and initiate differentiation [5]. Further analysis into the expression levels of cell cycle-related genes, such as *Cyclin*, *CDK*, and *CDKI* [39], at 40 days revealed their upregulation with more than a 2-fold change after *SsMeox1* knockdown, suggesting enhanced cell cycle progression in growth-specific muscle stem cells. However, the precise mechanism by which *SsMeox1* regulates the proliferation state of these stem cells remains elusive.

As a homeobox transcription factor, *Meox1* can regulate the expression of target genes by binding to their transcriptional regulatory regions [40]. *Ccnb1*, a key regulator of the cell cycle, plays a pivotal role in initiating mitosis [41]. Knockdown of *Ccnb1* can arrest the cell cycle at the G2 phase [42]. Intriguingly, *Meox1* can bind to the transcriptional initiation site of *Ccnb1* and suppress its expression. Overexpression of *Meox1* mimics the effect of *Ccnb1* knockdown, further supporting this regulatory mechanism [43]. Transcriptome analysis confirmed the upregulation of *Ccnb1* following *SsMeox1* knockdown. Dual-luciferase reporter assays demonstrated that *SsMeox1* directly inhibits *Ccnb1* expression, as in previous reports [5,43,44]. Thus, the knockdown of *SsMeox1*, followed by the upregulation of *Ccnb1*, promotes the cell cycle progression of growth-specific stem cells. However, the absence of *SsMeox1* disrupts the self-renewal capacity of these stem cells and accelerates the depletion of the stem cell pool, ultimately leading to growth defects.

## 4. Materials and Methods

### 4.1. Animal Materials

Embryos used for WISH at various developmental stages, including blastula, gastrula, somite, and pre-hatching, were collected from three-year-old female ovaries of *S. schlegelii*, which were collected from a deep-sea cage in Qingdao, Shandong Province, China. The juveniles used in the knockdown experiment came from a mixed cohort of offspring derived from ten female *S. schlegelii* broodstock, which was cultured in Zhongfu Hatchery Co., Ltd., Weihai, China. Muscle samples were taken from 0.5-year-old and 2.5-year-old individuals cultured in deep-sea cages in Qingdao, China. All the above groups had at least 3 individual replicate samples. Individuals less than one month of age were collected as a whole, and for older individuals, only muscle below the first dorsal fin was taken. Half of the samples were stored in liquid nitrogen for RNA extraction, and the other half were fixed in 4% paraformaldehyde (PFA) at 4 °C overnight, followed by methanol gradient dehydration.

### 4.2. Predicted 3D Protein Structure of SsMeox1

The three-dimensional (3D) structure of the SsMeox1 protein was constructed utilizing the SWISS-MODEL (https://swissmodel.expasy.org/, accessed on 23 March 2022), following the procedural guidelines outlined on the platform. 

### 4.3. ISH and WISH

One-month-old juvenile fish (body length: 24.8 ± 2.3 mm) and embryos were utilized for the ISH and WISH experiments. The probes specific to *SsMeox1* were amplified from cDNA using the primers listed in Table 1. The ISH probes of the *SsMeox1* gene were synthesized using a DIG RNA labelling kit (Roche, Mannheim, Germany). The ISH and WISH procedures followed previously established procedures [45]. The results of the ISH and WISH experiments were photographed using a Nikon AZ100 multizoom microscope (Nikon Corporation, Tokyo, Japan).

### 4.4. SsMeox1 Knockdown in Juvenile S. schlegelii

The selection of the *SsMeox1* ORF sequence fragment for siRNA induction was predicted through siDirect (http://sidirect2.rnai.jp/ (accessed on 12 December 2020)) and DSIR (http://biodev.cea.fr/DSIR/DSIR.htmL (accessed on 12 December 2020)). The fragment exhibiting the highest number of high-scoring siRNAs was selected as the final amplified template, with the related primers listed in Table 1. The construction of the recombinant *SsMeox1*-L4440-HT115 strain followed the methodology outlined in previous studies [46]. Briefly, the selected *SsMeox1* fragment was subcloned into an L4440 plasmid to form the knockdown plasmid *SsMeox1*-L4440. Subsequently, the *SsMeox1*-L4440 plasmid was transformed into the HT115 strain and selected in the presence of tetracycline (25 mg/L) and ampicillin (100 mg/L). A monoclonal strain of *SsMeox1*-L4440-HT115 was picked up and propagated for the induction of *SsMeox1*-targeted dsRNA. 

The recombinant strain was cultured at 37 °C until the OD reached a range between 0.6–0.8 at a wavelength of 600 nm. Subsequently, IPTG (Sigma, St. Louis, MO, USA) was added to each tube at a final concentration of 0.5 mM, inducing the culture at 28 °C for 5 h. After centrifugation at 10,000 rpm for 5 min, pure bacterial precipitate was obtained, from which a small portion was utilized for RNA extraction, and the efficiency of dsRNA production was evaluated via electrophoresis. 

Juveniles at 10 days post-hatching were randomly assigned to two pools measuring 2 m × 2 m × 1 m, each containing about 1000 individuals. The *SsMeox1* knockdown group was fed a diet containing the induced *SsMeox1*-L4440-HT115 strain, while the control group received a diet containing the induced L4440-HT115 strain. The precipitate was thoroughly mixed with the fish bait and allowed to stand for 10 min before feeding, with 400 mL of induced bacterial liquid per pool per day. This treatment persisted for 60 days, followed by a normal diet for an additional 60 days. Thirty individuals were randomly selected from each group every 10 days for body length measurements. Concurrently, a subset of samples underwent histological analysis via HE staining, while others were sacrificed for RNA extraction. Gene expression analysis and statistical assessments were conducted as previously described [47], employing qRT-PCR. The muscle fiber cross-sectional area was manually calculated using Image Pro Plus version 6.0 software (Media Cybernetics, Silver Spring, MD, USA).

### 4.5. Transcriptome Analysis

Transcriptome sequencing was conducted on muscle samples obtained 40 days after *SsMeox1* knockdown with 3 knockdown samples and 3 control samples. RNA extraction, sequencing library construction, and Illumina sequencing were performed by Novogene Bioinformatics Technology Co., Ltd. Briefly, the total RNA of the six samples (three biological replications per group) was extracted using Invitrogen TRIzol (ThermoFisher, Waltham, MA, USA). The RNA concentration and quality were analyzed using a Fragment Analyzer 5400 (Agilent, Santa Clara, CA, USA). The NEBNext^®^ Ultra^TM^ RNA Library Prep Kit (NEB, Ipswich, MA, USA) was utilized for library preparation. The library quality was assessed using the Agilent Bioanalyzer 2100 system (Agilent, Santa Clara, CA, USA). These libraries were sequenced using the NovaSeq 6000 platform. Salmon version 0.7.2 was used for the quantification of the transcript counts [48]. The gene expression levels were standardized by the transcripts per kilobase million (TPM) [49]. The analysis of differentially expressed genes (DEGs) was performed using the R package DESeq2 [50], and the genes with |log2 FC| ≥ 1 and an adjusted *p*-value < 0.01 were assigned as DEGs. In addition, GO and KEGG enrichment analysis were performed by DAVID (https://david.ncifcrf.gov/ (accessed on 14 October 2021)), and the results were visualized by R Studio.

### 4.6. Statistics Analysis

All statistics analysis were performed using SPSS version 20 (IBM, Armonk, NY, USA). An independent t-test was applied to compare the differences between the *SsMeox1* knockdown group and the control group. A *p*-value < 0.05 was considered statistically significant.

### 4.7. Dual-Fluorescein Reporter Gene Analysis

The JASPAR website (http://jaspar.genereg.net/ (accessed on 10 January 2022)) was utilized to predict the potential binding capacity of *SsMeox1* on the promoter regions of the target genes. Subsequently, promoter regions containing the predicted binding sites were amplified and subcloned into the pGL3-basic plasmid (Yeasen, Shanghai, China) to function as luciferase reporter vectors. The full-length sequence of *SsMeox1* ORF was amplified and subcloned into the pcDNA3.1 plasmid to serve as an overexpression vector. The primers used are listed in Table 1. Next, HeLa cells were plated into 24-well plate and cultured for 24 h at 37 °C, 5% CO_2_. Then, the cells were co-transfected with the corresponding promotor pGL3-basic and *SsMeox1*-pcDNA3.1, along with a Renilla plasmid, which works as an internal control for firefly luciferase activity, followed by a 24 h incubation period. In the control group, *SsMeox1*-pcDNA3.1 was replaced by an empty vector. The transcriptional activity of the target genes was quantified using a fluorescence intensity assay. Dual-fluorescein reporter gene analysis was conducted using a dual-luciferase assay system kit (Promega, Madison, WI, USA), adhering to the manufacturer’s instructions.

## 5. Conclusions

In summary, our study uncovers the crucial role of *SsMeox1* in regulating muscle growth in the large teleost fish, *S. schlegelii*. We identified *SsMeox1* as a marker for growth-specific muscle stem cells and demonstrated its essential function in maintaining the stem cell pool and promoting normal muscle growth. *SsMeox1* knockdown led to significant growth retardation, particularly affecting muscle fiber hypertrophy and leading to a significant reduction in body length after 60 days post-treatment. Further, the knockdown of *SsMeox1* can directly upregulate the expression of *Ccnb1* and subsequently promote the cell cycle of growth-specific stem cells, which ultimately led to stem cell pool exhaustion and growth defects. Our findings provide novel insights into the molecular mechanisms of muscle growth in large teleost fish, highlight the conserved role of *Meox1* across teleost fish species, and lay the foundation for future studies on sexual dimorphism in muscle growth and strategies for enhancing aquaculture production.

## Figures and Tables

**Figure 1 ijms-25-04871-f001:**
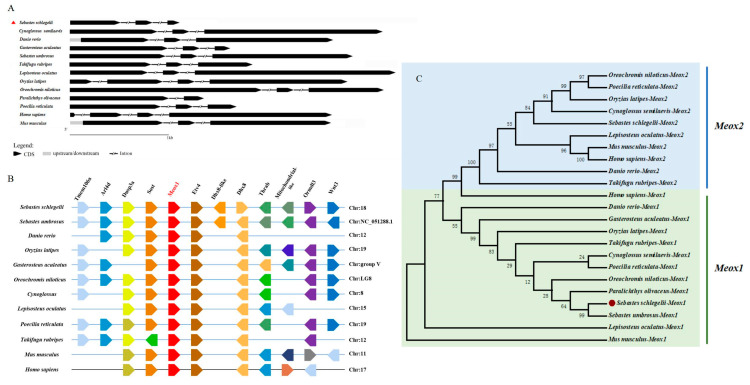
Characterization of *SsMeox1* in *S. schlegelii*. (**A**) Exon/intron structure of *SsMeox1* gene. Black arrows represent exons, and straight lines represent introns. (**B**) Collinearity analysis of *Meox1* genes in the genomes of diverse species. (**C**) Phylogenetic analysis of *Meox1* genes and *Meox2* genes in mammals and teleosts. Numbers displayed on the branch represent bootstrap value. Accession numbers are listed in Appendix A. The red dot marks the *SsMeox1* gene.

**Figure 2 ijms-25-04871-f002:**
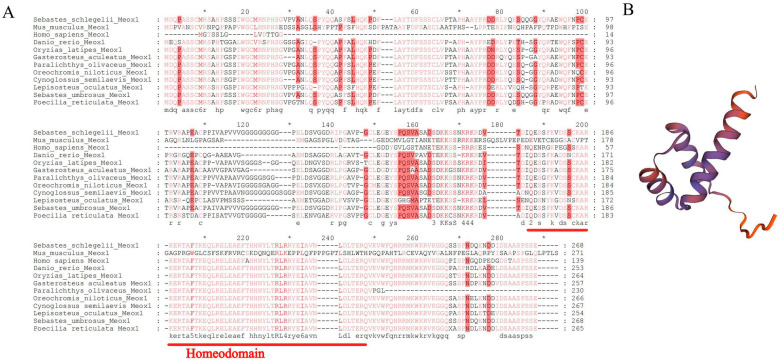
Multiple sequence alignment and predicted 3D structure of SsMeox1 protein. (**A**) The amino acid sequence alignment of *Meox1* genes in bony fish and mammals. The homeobox domain is underlined in red. The accession numbers and species names are listed in Appendix A. (**B**) The predicted 3D structure of SsMeox1 protein.

**Figure 3 ijms-25-04871-f003:**
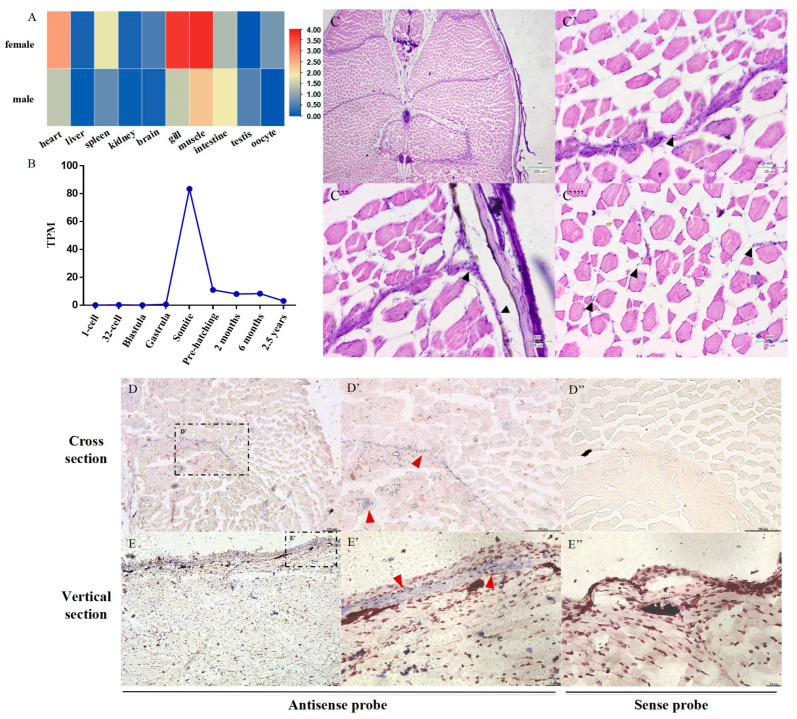
Expression patterns of *SsMeox1* and identification of growth-specific muscle stem cells. (**A**) Expression patterns of *SsMeox1* in different tissues (*n* = 3) based on TPM value. (**B**) Expression patterns of *SsMeox1* in different development periods (*n* = 3). The expression levels at 2 months, 6 months, and 2.5 years represent the expression levels of *SsMeox1* in individual muscles. The remaining periods represent the expression levels in entire embryos. (**C**) HE staining of cross sections of 3-month-old *S. schlegelii* (scale bars, 202 μm). The muscular septum (**C′**), the lateral ECL region (**C″**), and the muscle fiber region (**C′″**) are local magnifications of the whole cross section (scale bars, 25 μm). Black arrowheads mark growth-specific muscle stem cells. (**D**,**D′**) Spatial expression of *SsMeox1* in cross section of one-month-old *S. schlegelii*. (**E**,**E′**) Spatial expression of *SsMeox1* in vertical section of one-month-old *S. schlegelii*. (**D″**,**E″**) Sections labeled with sense probe were used as negative controls. The red arrows indicate positive signals (scale bars, 100 μm (**D**–**D″**), 20 μm (**E**–**E″**)).

**Figure 4 ijms-25-04871-f004:**
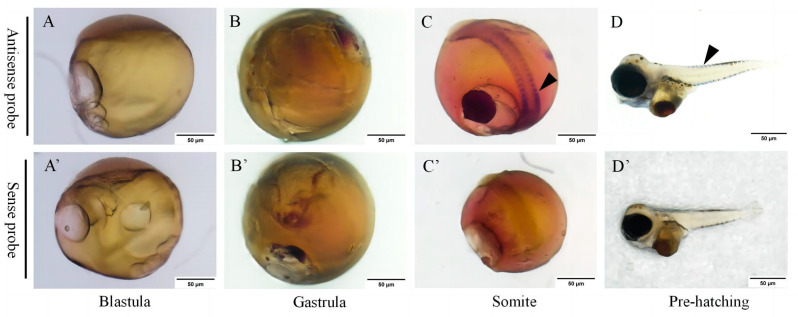
Expression patterns of *SsMeox1* at different embryonic stages: (**A**,**A′**) blastula, (**B**,**B′**) gastrula, (**C**,**C′**) somite, (**D**,**D′**) pre-hatching. (**A**–**D**) Embryos were labeled with antisense probe and positive signals were stained with blue. (**A′**–**D′**) Embryos labeled with sense probe were used as negative control (scale bars, 50 µm). The black arrows indicate positive signals.

**Figure 5 ijms-25-04871-f005:**
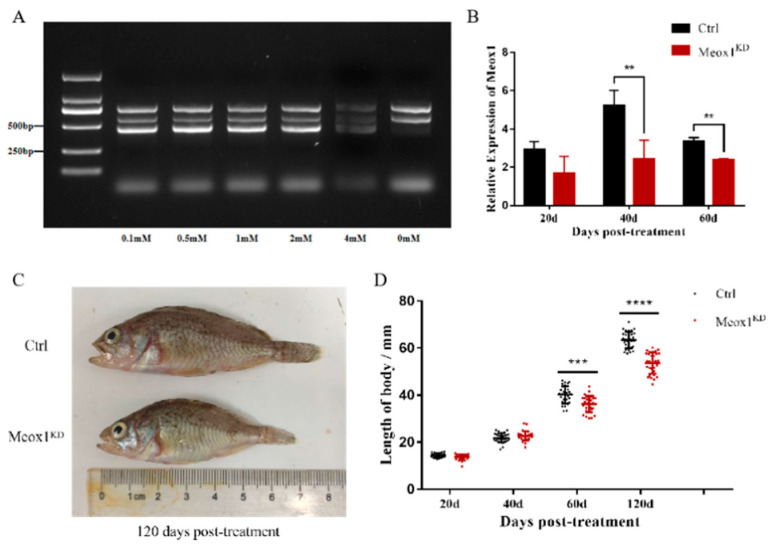
Determination of *SsMeox1* knockdown efficiency and the body length difference after *SsMeox1* knockdown. (**A**) Induction efficiency of double-stranded RNA at different IPTG concentrations. (**B**) The expression level of *SsMeox1* in knockdown group (*n* = 4) and control group (*n* = 4) at different treatment times by qRT-PCR. (**C**) Individual size of *SsMeox1* knockdown group and control group after 120 days of treatment. (**D**) Body length of the *SsMeox1* knockdown group (*n* = 30) and the control group (*n* = 30) at different treatment times. The black dots represent individuals in the control group, and the red dots represent individuals in the knockdown group; ** *p*-value < 0.01, *** *p*-value < 0.001, **** *p*-value < 0.0001.

**Figure 6 ijms-25-04871-f006:**
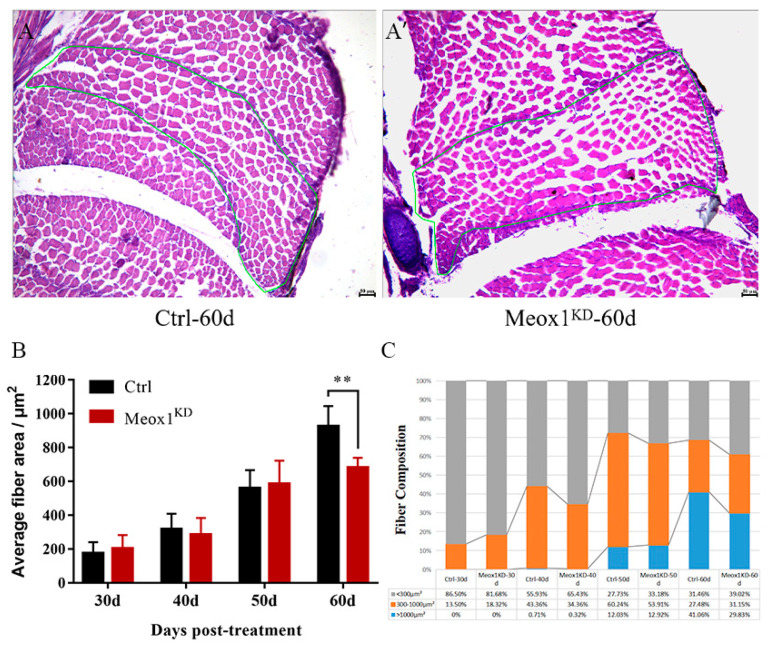
A comparison of muscle fiber cross-sectional areas between the control group and the knockdown group after treatment. (**A**,**A′**) HE staining of individual cross sections after 60 days of knockdown treatment. All muscle fibers in the green line were used to measure the cross-sectional area. (**B**) Average fiber area after knockdown treatment (*n* = 4). (**C**) Fiber-type composition after knockdown treatment (*n* = 4); ** *p*-value < 0.01.

**Figure 7 ijms-25-04871-f007:**
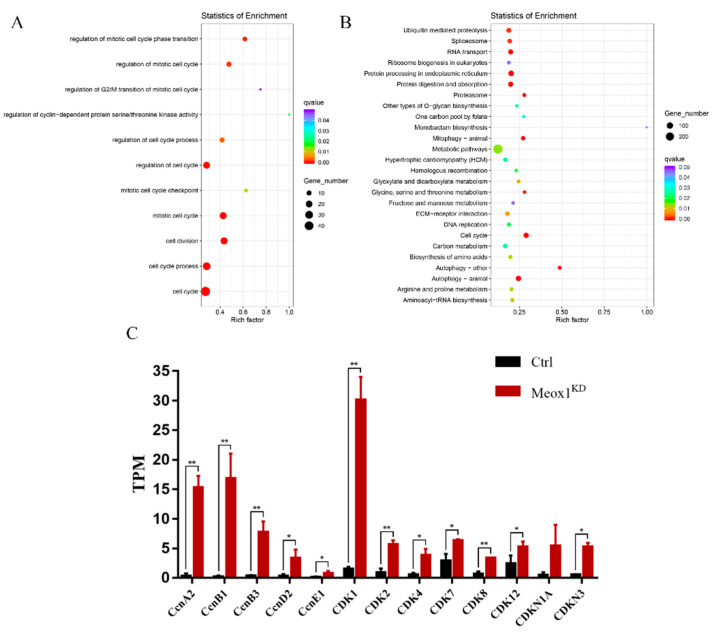
DEGs analysis of muscle transcriptome after knockdown treatment for 40 days. (**A**) Terms directly related to cell cycle progression by GO enrichment analysis. Terms with *q*-value < 0.05 are shown. (**B**) Pathways enriched by KEGG enrichment analysis. Pathways with *q*-value < 0.05 are shown. (**C**) Expression levels of all *Cyclin*, *CDK*, and *CDKI* family members in DEGs; * *p*-value < 0.05, ** *p*-value < 0.01.

**Figure 8 ijms-25-04871-f008:**
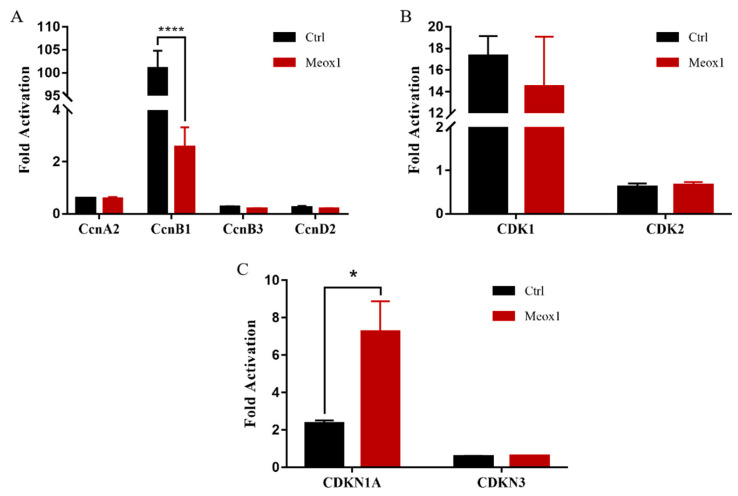
The downstream target genes of *SsMeox1* were screened using the dual-luciferase reporter system. (**A**) Regulation of *SsMeox1* on cyclin genes revealed by the dual-luciferase reporter system. (**B**) Regulation of *SsMeox1* on *CDK* genes revealed by the dual-luciferase reporter system. (**C**) Regulation of *SsMeox1* on *CDKI* genes revealed by the dual-luciferase reporter system. The ordinate value represents the ratio of firefly fluorescence to Renilla fluorescence; * *p*-value < 0.05, **** *p*-value < 0.0001.

**Table 1 ijms-25-04871-t001:** Primers used in this study.

Primer Name	Sequence (5′-3′)	Usage
Ssc-Meox1-ISH-SP6-fw	ATTTAGGTGACACTATAGAAGAGTGACACGGAGAGAGACT	ISH/WISH
Ssc-Meox1-ISH-T7-rv	TAATACGACTCACTATAGGGAGACTTATGGCTTGGCAACAC	ISH/WISH
Ssc-Meox1-qPCR-fw	AGTTCACCCACCACAACTAC	qPCR
Ssc-Meox1-qPCR-rv	GAGGCTGCTGAGTCAATGT	qPCR
CcnF-luc-hindIII-fw	CCCAAGCTTGGGCAGCACCACAAATGTGCCACC	DLR
CcnF-luc-kpnI-rv	CGGGGTACCCCGTGCCAAGCTGACATCATGCCA	DLR
CcnB3-luc-kpnI-fw	CGGGGTACCCCGTAGCAAATTGTACTGTAAATG	DLR
CcnB3-luc-hindIII-rv	CCCAAGCTTGGGAACTGATGTGCTGCACAGAAT	DLR
CDK1-luc-hindIII-fw	CCCAAGCTTGGGAGTTTCTAGTGGATTGAGCTG	DLR
CDK1-luc-kpnI-rv	CGGGGTACCCCGCATTCAGCGGCAGTCTGAGTA	DLR
CcnD2-luc-kpnI-fw	CGGGGTACCCCGCCTCACGAGGTGTCACTTGCT	DLR
CcnD2-luc-hindIII-rv	CCCAAGCTTGGGTTTTAAGCGGATTTACCGACG	DLR
CDK2-luc-hindIII-fw	CCCAAGCTTGGGGTAGACATTTCTTGTGCCATA	DLR
CDK2-luc-kpnI-rv	CGGGGTACCCCGAAGCTGTGTGGCTGACAGTTG	DLR
CcnA2-luc-kpnI-fw	CGGGGTACCCCGGGACATTCTGCATAATGATTA	DLR
CcnA2-luc-hindIII-rv	CCCAAGCTTGGGAAAAGTAAAAGTCCTACATTC	DLR
CcnB1-luc-kpnI-fw	CGGGGTACCCCGCCACTTCTTTCCACCACAAG	DLR
CcnB1-luc-hindIII-rv	CCCAAGCTTGGGTAATGTTGCAGCTGGTAAAGG	DLR
CDKN1A-luc-kpnI-fw	CGGGGTACCCCGGGGAAATGTAGGTGTGTTTCG	DLR
CDKN1A-luc-hindIII-rv	CCCAAGCTTGGGACTGGGCAGCTCTTTATAGG	DLR
CDKN3-luc-hindIII-fw	CCCAAGCTTGGGTCTTACCACCATGAACTATAC	DLR
CDKN3-luc-kpnI-rv	CGGGGTACCCCGGATTTACAACATGCTTCAGCT	DLR
Ssc-Meox1-KD-XbaⅠ-fw	GCTCTAGAGCGCTGGACAGTGTAGGGG	RNAi Vector
Ssc-Meox1-KD-HindIII-rv	CCCAAGCTTGGGAGGCTGCTGAGTCAATG	RNAi Vector

## Data Availability

The transcriptome data were submitted to the NCBI Sequence Read Archive (SRA) under the project number PRJNA1094898.

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
