# Peer review of "Regulatory Role of Meox1 in Muscle Growth of Sebastes schlegelii"

_ijms, 2024, doi:10.3390/ijms25094871_

Round 1

Reviewer 1 Report

Comments and Suggestions for Authors

IJMS - 2976452

Regulatory role of Meox1 in muscle growth of Sebastes schlege-lii 

Weihao Song#1, Xiaotong Liu#1, Kejia Huang1, Jie Qi1, Yan He*1 4

General comments on the manuscript are as follows:

1.     Abstract: The abstract is quite well prepared and suggestively. It presents the results of the research. Paper conclusions should be enforced.

2.     Introduction:

Data on Sebastes schlegelii importance and scientific elements that form the basis of the work are presented in a concise and clear manner in the introduction. As regards the scientific names, corrections must be made in all paper. (e.g Sebastes schlegelii)  

3.     Materials and methods Chapter:

The Materials and methods chapter should be placed in the third place not in 4 one. Correction must be done. The Materials and methods are structured in seven good developed subchapters. All the mandatory data of this chapter and many elements that are relevant for the observations and discussions chapter are presented. Description of methods and materials must be a very little restructured so that they form the specific Materials and methods Chapter. The seven subchapters stated in the paper, presents these elements less clearly.   4.     Results Chapter The Results Chapter (should be in 4 paper position, not in 2), comprises eight good developed subchapters. The chapter is properly structured and presented. Part of the results presented in the materials and methods chapter can be taken over and outlined in this chapter   5.     Discussions Chapter The Discussion Chapter should be in 5 paper position, not in 3 one. The chapter is properly structured and presented.   6.     Conclusions Chapter The Conclusions Chapter should be in 6 paper position, not in 5 one. For a better and clear understanding, the Conclusion Chapter should be slightly revised in order to emphasize the presentation of the results of the research undertaken based on the all relevant observations.  

References:

References are appropriate.

The paper is balanced with technical and scientific data. 

Reviewer 2 Report

Comments and Suggestions for Authors

The study primarily investigates the role of the Meox1 gene in muscle growth in Sebastes schlegelii, a large teleost fish. Meox1 is recognized as a critical transcription factor essential in embryogenesis and muscle development across various species. This research focuses on the expression and function of SsMeox1 and explores its regulatory impact on muscle growth, particularly concerning growth-specific muscle stem cells. Key findings from the study include: (1) Identification of SsMeox1 as a marker gene for growth-specific muscle stem cells, primarily located in the external cell layer and myosepta of muscle tissues; (2) Development and application of a Meox1 knockdown system in S. schlegelii, which demonstrated that the knockdown significantly impaired muscle growth and body length; (3) Phenotypic and transcriptomic analyses revealed that SsMeox1 regulates cell cycle progression by suppressing the expression of Ccnb1, maintaining the stability of the muscle stem cell pool crucial for continuous muscle growth. This research provides significant insights into the molecular mechanisms underpinning indeterminate muscle growth in large teleosts, offering a foundation for potential genetic and pharmaceutical interventions aimed at enhancing muscle growth in commercial fish species. I recommend accepting this manuscript, provided that the authors make revisions according to the suggestions I have provided.

 line 65, the term "knockdown" is used without consistent clarification whether it refers to partial or complete gene silencing, which could lead to confusion about the extent of gene inactivation achieved in the experiments;

Figure 1 and 2, the resolution is too poor to allow readers to see it clearly;

 line 157, the knockdown approach requires further elaboration to facilitate reproducibility by readers and fellow researchers;

 line 278, the paper states "significant growth retardation" without quantitatively defining what constitutes significant growth retardation, which makes the statement vague and subjective; The expression "significant upregulation" is used without providing specific fold-change or statistical values, which can mislead the reader regarding the true impact of the findings . 

line 454, The format of the references is inconsistent; please standardize the citation format according to the guidelines of the journal;
